# Critical Factors Affecting the Prevalence of *Staphylococcus aureus* and Staphylococcal Enterotoxins in Raw Milk Cheese in the Alpine Region of Austria, Italy, and Switzerland

**DOI:** 10.3390/foods14132176

**Published:** 2025-06-21

**Authors:** Thomas F. H. Berger, Milena Brasca, Margaretha Buchner, Ueli Bütikofer, Bianca Castiglioni, Paola Cremonesi, Frieda Eliskases-Lechner, Lena Fritsch, Stefano Morandi, Livia Schwendimann

**Affiliations:** 1Agroscope, 3003 Berne-Liebefeld, Switzerland; ueli.buetikofer@agroscope.admin.ch (U.B.); lena.fritsch@agroscope.admin.ch (L.F.); livia.schwendimann@sqts.ch (L.S.); 2Institute of Sciences of Food Production, National Research Council, 20133 Milan, Italy; milena.brasca@cnr.it (M.B.); stefano.morandi@ispa.cnr.it (S.M.); 3Federal College and Research Centre for Agriculture and Nutrition, Food- and Biotechnology Tyrol, Rotholz 50, 6200 Strass im Zillertal, Austria; margaretha.buchner@hblfa-tirol.at (M.B.); frieda.eliskases-lechner@hblfa-tirol.at (F.E.-L.); 4Institute of Agricultural Biology and Biotechnology, National Research Council, 20133 Milan, Italy; bianca.castiglioni@ibba.cnr.it (B.C.); cremonesi@ibba.cnr.it (P.C.)

**Keywords:** raw milk cheese, Alpine region, *Staphylococcus aureus* prevalence, enterotoxins, critical production factors, food safety, hurdle index

## Abstract

In the Alpine region of Austria, Italy, and Switzerland, transhumance is widespread and the production of local traditional dairy products during summer is important. Raw milk cheeses are produced according to traditional recipes, using hurdles as a technique to guarantee food safety. In the present study, we aim to provide an overview of *S. aureus* and its enterotoxins in raw milk cheeses, identify the key parameters responsible for the enterotoxin production, and identify ways to improve food safety. The results demonstrate that safe artisanal raw milk cheese production is achievable under elementary conditions by applying effective hurdles, including high scalding temperatures or thermization, quality starter cultures, and robust milk quality management. The hurdle index (HI), which we introduce in this paper, is a promising tool for assessing and improving safety in raw milk cheese production.

## 1. Introduction

Bovine mastitis (bovine intramammary infection, IMI), which is caused by pathogens, is the most important and costly disease among dairy cows worldwide [1], and it causes severe losses in dairy cattle herds via reduced milk yield, the reduced suitability of the milk for dairy product manufacturing, antibiotic treatment costs, and the culling of animals in cases of treatment failure [2]. *Staphylococcus aureus* (*S. aureus*) is one of the most common food pathogens isolated from milk and dairy products [3]. This microorganism is commonly found in a wide variety of mammals and mainly transferred to food by clinical/subclinical staphylococcal mastitis and environmental contamination during the handling and processing of raw milk [4].

Via genotyping with ribosomal spacer PCR [5], more than 100 genotypes and variants of *S. aureus* have been identified in milk [6]. Different genotype (GT) variants were then combined to form genotype clusters (CLs). CLB (Cluster B, containing GTBs), CLC, and CLR were found to be the most common genotypes in a European study [7], with CLB being restricted to central Europe, especially in Switzerland and bordering countries, whereas CLC and CLR were found in almost every country. A worldwide collection of *S. aureus* isolates [8] revealed the wide variety of *S. aureus* genotypes in dairy cattle, with CLR and CLC being the most frequent. The remaining genotypes were rare and accounted for a minority of all isolates [5,7,8]. In the Alpine region, GTBs and GTCs are often the most prevalent genotypes, with GTBs being genotypes of a contagious mastitis-causing pathogen that can easily spread among cows [5,9,10,11]. Regarding contagious *S. aureus* mastitis, it has been shown that the sanitation of affected herds [12] and entire regions [13] is possible and that the economic losses described above can be avoided. In addition, the potential food safety risk can also be decreased in this way, as only raw milk products that are produced in conformity with Commission Regulation (EC) No. 2073/2005 [14] will reach consumers.

*S. aureus* is a versatile pathogen producing a variety of exotoxins, including Staphylococcal Enterotoxins (SEs), which are associated with food-borne intoxications, specifically Staphylococcal Foodborne Poisoning (SFP) [15]. First described in 1959, SEs and Staphylococcal Enterotoxin-like proteins (SEls) attack intestinal cells, causing gastroenteritis with symptoms such as vomiting and diarrhoea [16]. These proteins are resilient; resistant to heat, digestive proteinases, irradiation, and denaturing agents; and stable across a wide pH range [17,18,19]. To date, 26 SEs have been identified, with new types constantly emerging due to advancements in molecular and genetic characterization techniques. The International Nomenclature Committee for Staphylococcal Superantigens (INCSS) proposed a standard naming convention in 2004, highlighting the relevance to food poisoning (emetic activity). Toxins are classified SEs if they demonstrate emetic activity in primate models or as SEls if their emetic potential is not established [20].

According to the European Food Safety Authority (EFSA), SEs are the second most important cause of foodborne outbreaks due to bacterial toxins in humans [21]. The literature describes a wide range of outbreaks involving the classical enterotoxins (SEA–SEE). In particular, SEA is known to cause large outbreaks, including some with over 10,000 cases [17,22]. SEs from *S. aureus* transmitted from humans (human strains) or from dairy animals (mastitis) to food during food processing are the two main sources of SFP outbreaks [19]. Because methods such as VIDAS SET2 and Ridascreen are commercially available and can detect the presence of toxins SEA–SEE, related outbreaks can readily be detected and described [23]. However, often, the implicated strains also carry non-classical enterotoxin genes. The lack of commercially available detection methods, such as ELISA, for non-classical enterotoxins complicates their identification, making it difficult to fully document and confirm their involvement in outbreaks [24,25].

The studies of Graber et al. [26] and Cosandey et al. [7] demonstrated a strong association between genotypes and virulence gene patterns based on the presence of various enterotoxin genes and gene polymorphisms. In particular, enterotoxin genes such as *sea*, *sed*, *sej*, and *ser* characterize GTBs, which can persist along the food chain, becoming responsible for foodborne outbreaks involving raw milk cheeses, as previously described by Hummerjohann et al. [27], Johler et al. [28], and Kümmel et al. [29]. Fresh, soft, and semi-hard cheeses made from raw milk are especially a concern due to their association with SFP outbreaks.

Cases implicating non-classical enterotoxins have been reported in recent years. Notably, staphylococcal enterotoxin H (SEH) and the enterotoxin gene cluster (*egc*) enterotoxins appear to be the most involved non-classical enterotoxins. Two outbreaks involving SEH were detected in coagulase-positive staphylococci (CPS) strains harboring the *seh* gene [30,31]. For *egc* enterotoxins, there is evidence of their involvement in multiple outbreaks, but due to the lack of suitable detection methods, the involved SEs were never analysed [32,33].

During cheese making, various stress factors (pH, redox potential, and salt concentration) and technological parameters (scalding and ripening temperatures and competition from starter cultures) can influence *S. aureus* growth and enterotoxin production [34,35,36]. Duquenne et al. [34] showed that temperature and time at the beginning of the cheese-making process were key parameters in SE production, while Schwendimann et al. [6] highlighted the fact that enterotoxin formation was also related to the type of *S. aureus* present in milk.

To reduce the number of SEs intoxications, European legislation has defined microbiological criteria for foodstuffs, such as CPS enumeration and enterotoxin recognition. In particular, SE detection must be performed when the CPS count exceeds 10^5^ colony-forming units per gram. If SEs are present in a 25-g test portion, the food is considered unsafe and must not be placed on the market or must be withdrawn from the market (article 14) [37].

In the geographic zone under consideration in this study (the Alpine region of Austria, Italy and Switzerland), transhumance is widespread. In the European Alpine case, transhumance is the summer transfer of livestock from lowland to highland pastures, a traditional practice that is considered as an integral part of the mountain ecosystem [38,39]. Transhumance is probably one of the oldest forms of pasturing and can be traced back, based on written sources, to the Middle Ages; based on archaeological findings, to the Bronze Age (2200–800 BC); and based on palynological studies, to the use of high-altitudes pastures beginning in 4500 BC [40,41]. The oldest evidence of sheep farming in the Southern Alps was found in France dates back to over 8000 years [42]. Transhumance farming has allowed individuals to survive as mountain farmers despite often-difficult conditions [43]. Its practice provides additional forage supply and plays a role in the preservation and shaping of the Alpine landscape, biodiversity, natural habitats, and conservation of local traditional dairy products [38,43,44,45]. During the eighteenth and nineteenth centuries, the mountains became a central object of romanticism. Thus, transhumance often contributes to tourism and recreational activities [43,45]. Milk obtained at Alpine pastures is mostly destined to produce high-value local cheeses, some of which are linked to their geographical origin with AOP/POD (Product of Designated Origin) labels (AOP stands for “Appellation d’Origine Protégée” and means “protected designation of origin” (POD) in English. It is a label that guarantees that the origin, production and processing of a product takes place in a specific, legally defined region. This means that everything, from the raw materials to the end product, must come from this area, using traditional methods and processes. It is a seal of quality that ensures that a product is authentically linked to its region of origin and guarantees its traditional production and quality). Dairy products in general [46] and those from Alpine pastures in particular are presumed to be healthy, due to their favorable fatty acid profile, and they are positively perceived by consumers [45]. On the Alps, the requirements regarding quality, hygiene, and protection against deception regarding the cheeses produced are identical to those in valley farms. However, the requirements have been partially simplified, as is possible, under food law, for micro-enterprises [47,48]. Because many traditional cheeses are produced according to traditional recipes by individuals with a great deal of experience, it can be assumed that food safety has been historically proven by applying hurdle technologies [49]. However, changes in personnel, the suppliers of starter cultures or other practices, and climate can lead to food safety problems [50,51].

Therefore, the aims of the present study were to (i) provide a broad overview of *S. aureus* and its enterotoxins in all types of artisanal raw milk cheeses in the Alpine regions in Europe (Austria, Italy, and Switzerland), (ii) identify the key parameters responsible for enterotoxin production, and (iii) identify ways to improve food safety.

## 2. Materials and Methods

### 2.1. Sampling and Sample Collection

Manufacturers of raw milk (Phosphatase-positive milk) cheese on the Alps, specifically various regions (See also the region groups in Appendix A) in Austria (Tirol-Kärnten AT1, Vorarlberg AT2), Italy (Bergamo IT1, Brescia IT2, Verbano-Cusio-Ossola IT3), and Switzerland (Western Switzerland CH1, Central/Eastern Switzerland CH2, Ticino CH3), were invited to participate voluntarily in the study and submit cheese samples (Appendix A and Figure 1) at the beginning of the 2021 Alpine season (June or July). The manufacturer performed sampling according to a detailed protocol, cutting off a segment (height at least 3 cm) of molded cheese wheel before or directly after brining (Figure 2). Differences in the distribution of samples were accepted and the types of cheeses produced in the various regions and in what quantities or tonnage were not determined. Except for hard cheese samples, this is the time during the manufacturing process when the number of CPS is expected to be highest. After the preparation of the test sample, the remaining sample was frozen (−18 °C) for the detection of SEs. A total of 126 cheese samples were analysed, including Swiss fresh cheeses (FC, *n* = 6); Austrian hard cheeses (HC, *n* = 19); and semi-hard cheeses from Switzerland (SHC, *n* = 58), Italy (*n* = 30) and Austria (*n* = 13). The Italian and Austrian cheeses were made from cow milk only, whereas some of Swiss cheeses were produced with goat’s milk (*n* = 21) or a mixture of cow and goat milk (*n* = 2).

### 2.2. Collection of Data on Milk Processing and the Manufacturing Conditions for Cheese

Each manufacturer participated in a questionnaire survey on manufacturing conditions. The queried data included cheese types, raw milk type (cow, sheep, or goat), milk storage (time and temperature (The temperatures given in Appendix A exceed, in some cases, those specified in Regulation (EC) No. 853/2004 [52]. However, it is also specified in Regulation (EC) No. 853/2004 that food business operators need not comply with the temperature requirements laid down in points 2 and 3 if the milk meets the criteria provided for in Part III and either (a) the milk is processed within two hours of milking or (b) a higher temperature is necessary for technological reasons related to the manufacture of certain dairy products and the competent authority so authorises)), culture type (bulk or direct starter and commercial/natural/no starter), point of culture addition, maximum temperature/time exposition until pressing, and pH two h after molding and/or before brining (Appendix A).

### 2.3. Enumeration and Isolation of CPS

To ensure the enumeration of CPS within 24 h after sampling, each participating institute performed the microbiological analyses in their respective laboratories (AT: HBLFA Tirol, IT: ISPA CNR, CH: Agroscope). CPS counts were determined according to ISO 6888-2:1999 [53] on Baird-Parker-RPF-Agar (bioMérieux, Marcy-l’Étoile, France; Biolife Italiana srl, Milan, Italy). Five typical colonies per plate (preferably having different appearances) from samples with CPS counts >1000 cfu/g were isolated and stored for further investigation.

### 2.4. CPS Strains Identification

Matrix-assisted laser desorption ionization–time of flight mass spectrometry (MALDI-TOF MS) was used to identify the CPS strains isolated from cheese samples at the microbiology laboratory of the Department of Veterinary Medicine and Animal Science (University of Milan, Italy) and at Agroscope (Microbiological Food Safety group). Fresh cultures of CPS isolates were cultured on BHI agar plates, and cell material from an isolated colony was deposited on the target plate using a toothpick. Samples were overlaid with 1 μL of α-cyano-4-hydroxycinnamic acid in 50% acetonitrile with 2.5% trifluoroacetic acid (Bruker Daltonik GmbH, Bremen, Germany). The spectra were acquired with a microFlex™ mass spectrometer (Bruker Daltonik GmbH, Bremen, Germany) in the positive mode. Bacterial test standard (Bruker Daltonik GmbH) was used for instrument calibration. Spectra were automatically interpreted using the database MBT Compass^®^ 4.1. A log (score) of 1.7 was the threshold for genus-level identification, and a log (score) of 2.0 was the threshold for species-level identification.

### 2.5. CPS Genotyping

Nucleic acid extraction from fresh cultures and genotyping based on the PCR amplification of the 16S–23S rRNA intergenic spacer region were performed as described by Fournier et al. [5].

### 2.6. SE Analysis in Cheese Samples

SEs were detected in the cheese samples using the ISO 19020:2017 [54] method and an immunoenzymatic assay (Vidas SET2, bioMérieux, Marcy-l’Étoile, France).

### 2.7. Statistical Analysis

Data were statistically analyzed using SYSTAT for Windows Version 13.0 (Systat Software, Richmond, CA, USA). A non-parametric, robust Kruskal–Wallis test was used to examine the influence of geographical and technical variables. Pairwise comparisons with the Conover–Inman test were performed to show statistically significant differences. Other statistical tests (a Wilcoxon Signed-Rank Test and the chi-squared test) were used, depending on the question to be answered. Differences were classified as statistically significant if *p* ≤ 0.05.

## 3. Results

### 3.1. General Findings

Of the 126 cheese samples included (Appendix A), 103 were made from cow milk, 21 from goat milk, and two from a cow-milk–goat-milk mixture. As this study focuses on cheeses made from raw milk, which face a high risk of *S. aureus* contamination, or milk that has undergone a lower heat treatment than pasteurization [52] (phosphatase-positive milk, which is typically referred to as thermized milk), samples no. 31 (SHC, CH1), 90 (SHC, CH3), and 92 (FC, CH3) were eliminated because they were pasteurized (Appendix A in bold italic and Figure 3). Thus, the final dataset was composed of 123 samples.

Figure 4 shows the distribution of the log(CPS) values. In 52% of the samples, *S. aureus* was not detectable (log(CPS) values < 0.5). In 32%, it was detectable, and log(CPS) values were between 0.5 and 5. In 16%, the log(CPS) values were between 5 and 6.5.

### 3.2. Differences Between Countries and Regions

As expected, there was a significant difference in log(CPS) values between countries and between some regions (Section 2.1, Appendix A) because of the vastly different cheese types sampled in these countries and regions and their effect on CPS counts, (e.g., AT1 [SHC] versus AT2 [HC] or CH1 [SHC] versus CH3 [FC]). Surprisingly, *S. aureus* sanitation in the Swiss region of Ticino, CH3, [13] did not show a clear effect. Significant differences for Ticino, CH3, were only observed for AT2 (HC) and, CH1 and CH2 (both SHC).

Because 98 of the 123 cheeses made from raw or thermized milk were SHC, significance testing was repeated on these samples only. AT1, CH1, and CH2 were significantly different from IT2 and IT3. Additionally, CH1 and CH2 were different from CH3, the sanitized region of Ticino. Like between the regions, there were also some significant differences between the SHC of the various countries, such as between IT and AT, as well as between IT and CH, but not between AT and CH.

### 3.3. Differences Between Types of Raw and Thermized Milk and Cheese Varieties

There was no significant difference in log(CPS) between the cheeses made from cow milk (*n* = 102) and goat milk (*n* = 19). It is worth noting that goat milk cheeses were free of *S. aureus* GTB and classical SE (Appendix A).

The three cheese types, HC (*n* = 19), SHC (*n* = 98), and FC (*n* = 6) were tested for significant differences in log(CPS). Significance was found between HC and FC, as well as between HC and SHC (Appendix A).

The differences in log(CPS) between the cheese varieties were significant (Appendix A), and they are visualized in Figure 5 (*n* = 123). Varieties that showed no significant differences were grouped into five partially overlapping clouds (in the shape of ovals; Cloud 1, Bergkäse; Cloud 2, Mutschli/Tomme; Cloud 3, Schnittkäse; Cloud 4, Tilsiter; and Cloud 5, Formagella). Characteristic values were calculated for each cloud (Appendix A). The differences between the clouds are because of the various hurdles used in cheese production, namely temperature; ripening, as a measure of water activity (a_w_) through drying; pH value; and competing microbial species. For each cloud, the median or mean values of the corresponding parameters were calculated (Appendix A); i.e., the maximum applied temperature in °C, the ripening time in weeks [if ranges were given, the corresponding mean value was used], the lowest given pH value [from “pH after 2 h”, “pH before salting”, or a pH of 7 was used as a worst-case scenario if no pH value was given] and a newly introduced value indicating the quality of the starter cultures used [flora value (FV), with a value of 1 for commercial cultures; 0.5 for natural cultures, as their quality is sometimes questionable; and 0.1 if no culture was used]). These values were based on significant differences in log(CPS) between culture type, which was recorded as their “commercial” or “no” (Appendix A, but not in “pH before salting”, and Appendix A). We therefore assume that the values given above are reasonable. In addition, a new hurdle index (HI) was calculated as a measure of hurdle effectiveness by multiplying the maximum temperature by FV and the ripening time and dividing it by pH (HI = temp(max) × FV × ripening/pH). This was done with the median or mean values for each cloud (Appendix A) but also for each sample if the required data were available (Appendix A). Figure 6 shows a diagram in which the log(CPS) values are plotted against HI. An HI of 35, chosen graphically from Figure 6, appears to be a reasonable basis for evaluating raw milk cheeses with respect to CPS or SE risk. After applying the HI to the clouds using their median or mean values, the cheeses in Clouds 1–3 appear to be potentially safe, while the cheeses in Clouds 4 and 5 are potentially risky. Both SE-positive cheeses were found in these last two clouds.

The testing of the different varieties for the significance of log(CPS) was repeated with SHC only (Appendix A; *n* = 98), without the HC Bergkäse and the FC Büscion but with one remaining SHC, Büscion di Capra. Again, varieties that showed no significant differences were grouped into five partially overlapping Clouds (Figure 7). Cloud 1, Bergkäse, was no longer necessary. Clouds 2, Mutschli/Tomme, and Cloud 3 Schnittkäse remained unchanged, whereas Cloud 4 lost Büscion di Capra and added Toma, Formagella and Formaggio di Alpe Ticinese AOP. Cloud 5 lost Büscion di Capra, Tilsiter and Toma. A new cloud, Cloud 6, was created, it consisted of Büscion di Capra, Schnittkäse, Minadur, and Raclette d’alpage. Characteristic data were again calculated for each cloud (Appendix A) in the form of median or mean values.

### 3.4. Differences Between Types of Culture

For the culture starter variable, significant differences in log(CPS) were calculated between bulk and direct (*n* = 107, bulk, *n* = 44; bulk + direct, *n* = 7; direct, *n* = 56). For the culture origin variable, these were calculated between commercial and no (*n* = 112 overall; “-” [nonresponsive], *n* = 2; for commercial, *n* = 95; natural, *n* = 8; and for “no”, *n* = 7), but not for the temperature tolerance of the cultures (*n* = 94 overall, for mesophilic [M], *n* = 12; for thermophilic [T], *n* = 20; and for meso-thermophilic [MT], *n* = 62] (Appendix A).

There were significant differences in pH before salting based on the temperature tolerance of the cultures (*n* = 38 between M and MT, and between M and T; for M, *n* = 5; for T, *n* = 5; and for MT, *n* = 28), but not based on culture starter or culture origin (Appendix A).

Regarding pH after 2 h, the bulk culture type had a range of from 5.15 to 6.52 (SE-positive sample no. 135 had pH 6.52; see also Appendix A), and regarding pH before salting it had a range of from 5.15 to 5.35 (no value for SE-positive sample no. 135). The log(CPS) values ranged from 0 to 5.431. Regarding pH after 2 h, the direct culture type had a range of from 5.21 to 6.54 (SE-positive sample no. 13 had a pH 6.42; see also Appendix A), and regarding pH before salting, values ranged from 4.2 to 5.84 (SE-positive sample no. 13 had a pH 5.84; see also Appendix A). The log(CPS) values ranged from 0 to >6 with several being >6.

The remaining pH data were as follows: bulk + direct culture starters had a pH-after-2h that ranged from 5.38 to 6.02 but a pH-before-salting range from 5.2 to 5.3. “No” had no data for pH after 2 h and a pH-before-salting range from 5.2 to 5.4, while natural culture starters had a pH-after-2-h range from 5.21 to 6.00 and a pH-before-salting range from 4.95 to 5.3.

Goat’s milk cheese producers (FC, *n* = 6; SHC, *n* = 13) applied a maximum temperature between 28 and 48 °C (*n* = 7) or between 49 and 67 °C (*n* = 8). One producer provided only a storage temperature, and three provided no temperatures at all. Starter type was either bulk (*n* = 1), direct (*n* = 15), or “no information” (*n* = 3). pH after 2 h was not given, and pH before salting was between 4.27 and 5.4 (*n* = 13) or not given (*n* = 6). log(CPS) was between 0 and 3 (*n* = 13) or between >3 and 4.792 (*n* = 6). No SE was found.

### 3.5. Calculation of Risk Groups Based on a Selected Number of Hurdle Indices

We introduced the HI as a tool with which to assess conditions in artisanal raw milk cheese production. To our knowledge, this is the first attempt to evaluate the effectiveness of hurdle technology. It is a promising approach because it assesses the given technological situation, but it requires active measurement data to determine whether the hurdles are effective. In the calculation of HI regarding SE risk, ripening time may play little or no role, as SEs are produced in the first hours of cheese production. However, the ripening time is important for the assessment of potential *S. aureus* contamination.

Using our dataset, an HI value of 35 was identified as a threshold for safe cheeses. Although this value will need to be validated using a larger number of data, it provides a simple and effective method for assessing the potential risk of the presence of SEs. In the cases of samples no. 135 and 13, the HI would have indicated a risky production environment, as producers applied maximum temperatures of less than 48 °C (SE-positive sample no. 135: 37 °C; sample no. 13: 43.6 °C). Assuming the starter cultures were functioning optimally (pH <5.7), using 5.6 as a pH value and an FV of 1, the corresponding HIs would have been 26.43 (no. 135; HI = 37 × 1 × 4/5.6) and 31.14 (no. 13; HI = 43.6 × 1 × 4/5.6). Using the measured pH values, the HIs were 22.70 and 29.86, respectively. In a case in which the maximum temperature was 49 °C (>48 °C), the HI would have been 35.

The data from Table 1 are used to further evaluate the HI, including the two SE-positive samples, 13 and 135, in bold.

Samples 78 and 79, in case (a), share similar characteristics in that pH plays a crucial role due to the low maximum temperature. In both samples, the HI is low and tends to overestimate the influence of temperature while underestimating the importance of pH, which is partially linked to the competing microbes and only partially accounted for by the FV. Sample 78 exhibits lower *S. aureus* counts (log(CPS)), likely due to its lower pH and possibly higher levels of competing microbes. In case (b), samples 1, 4, and 15 are characterized by critically low maximum temperatures, with FV values also being low and pH levels being below 5.7. For samples 13 and 135 (case (c)), both maximum temperature and pH as well as competitive microorganisms are critical factors. In contrast, samples 83 and 84 (case (d)) have maximum temperatures and pH levels that are within the safe ranges. However, the HI is underestimated due to an overestimation of the ripening time. In case (e), sample 11 has a pH below 5.7, but the maximum temperature still allows for the growth of *S. aureus*. Similarly, sample 129 (case (f)) has potentially dangerous maximum temperature, pH, and competitive microbe values, although the raw milk used in this case may have been free of *S. aureus*. For sample 43, in case (g), the maximum temperature is close to 48 °C, but pH and competitive flora remain critical, and the raw milk may also have been free of *S. aureus*. In case (h), sample 103 exhibits critical maximum temperature, pH, and competitive microbe values. However, in this case, the HI is overestimated due to an overestimation of ripening time. Sample 80, in case (i), shows safe maximum temperature and pH levels, and the HI appears to be reasonable. This sample, which is a mix of goat and cow milk, was likely highly contaminated with *S. aureus*. Similarly, sample 34 (case (j)) has safe maximum temperature and pH levels, with a reasonable HI. Finally, in case (k), samples 86 and 87 also have safe maximum temperature and pH levels, but the HI is overestimated due to an overestimation of ripening time. The milk used in these samples was likely highly contaminated with *S. aureus*.

The evaluation of these 17 selected cases led to the creation of six distinct risk groups, Group A includes cases in which maximum temperature is critical (<48 °C), pH is critical (>5.7), and the raw milk is contaminated. This group includes cases (b), (c), (d), and (h). Group B involves cases with a critical maximum temperature (<48 °C) and a critical pH (>5.7) but raw milk that is free of *S. aureus* (case (f)). Group C comprises cases with a critical maximum temperature (<48 °C) but a good pH (≤5.7), with raw milk contamination. This group includes cases (a) and (e) (if the temperature is appropriate, a critical pH should be as well). Group D includes cases with a good maximum temperature (≥48 °C) but a critical pH (>5.7) and raw milk that is free of *S. aureus* (case (g)); cases with a critical temperature and a good pH should also be possible. Group E consists of cases with a good maximum temperature (≥48 °C), a good pH (≤5.7), and contaminated raw milk. This group includes cases (i) and (k). Finally, Group F includes cases with a good maximum temperature (≥48 °C), a good pH (≤5.7), and raw milk free of *S. aureus*. Case (j) falls into this group.

## 4. Discussion

This study investigated *Staphylococcus aureus* (*S. aureus*) contamination and SE production in raw milk cheeses from the Alpine regions of Austria, Italy, and Switzerland. It focused on identifying the key factors influencing contamination and toxin production as well as methods via which to improve food safety in artisanal raw milk cheese production. Additionally, the study introduced the HI to assess the effectiveness of technological and environmental hurdles in ensuring safe production. The findings highlight the fact that safe artisanal raw milk cheese production is possible under elementary conditions if critical hurdles are effectively managed.

### 4.1. Prevalence of S. aureus and the Detection of SE in Cheese Samples

The study revealed key findings regarding the prevalence of *S. aureus* and the detection of SE in cheese samples. Overall, 52% of the cheese samples were entirely free of *S. aureus*. Meanwhile, 32% of the samples contained *S. aureus* counts (log(CPS)) below the legal limits, and 16% exceeded these limits (log(CPS >5)), with SEs being detected in 1.6% of samples.

As the samples were taken from the molded cheese wheel before or directly after brining, when, except for hard cheese, the number of CPS is expected to be highest, a comparison with published data is rather difficult. In many cases, authors refer to raw milk cheese, but do not provide any information on the process parameters (maximum temperature applied, type of culture, pH) which strongly influence the results. Milk quality and hygienic conditions are sometimes known, depending on the study design.

The analysis conducted by Zhang et al. [57] showed that South America had the highest prevalence rate of *S. aureus* in dairy products (49.03%), followed by Asia (19.11%), Africa (9.60%), Europe (7.01%), and North America (<3.00%), which probably reflects more the general habits regarding milk treatment (raw, thermized, pasteurized) and the effectiveness of hurdle techniques. According to the EFSA report [58] in 2022, SEs caused in the European Union 207 foodborne outbreaks with hospitalization, eight outbreaks being linked to milk and milk products. Outbreaks linked to the consumption of milk and milk products were reported by 12 Member States (Belgium, Finland, France, Germany, Hungary, Italy, Lithuania, Poland, Romania, Slovakia, Spain and Sweden) and two non-member states (Republic of North Macedonia and Serbia).

Hunt et al. [59] determined *S. aureus* in raw milk fresh and ripened cheese from four Irish cheesemakers. 11 of 16 fresh and none of two ripened cheeses were *S. aureus* positiv, log(CFU) values ranging from <2 to 4.146 for fresh cheeses and <1 for ripened cheese. In Scotland, Williams and Withers [60] examined 28 artisanal semi-hard and hard cheeses from small producers, made from cow’s, sheep’s or goat’s milk using raw and pasteurized milk. The ripening period ranged from a few days to around 12 months. Of the raw milk cheeses, 40% were *S. aureus*-positive and 33.3% of the cheeses made from pasteurized milk. The log(CFU) values ranged from 2 to >5. Jørgensen et al. [61] examined the presence of *S. aureus* on a Norwegian summer-farm facility and found log(CFU) values of 4.176 in the curd at pressing, 3.778 in the cheese after 7 days, and <1 after 10 week of ripening. The cheese was heated up to a maximum temperature of 39 °C, mesophilic and thermophilic starters were used, and the pH value after 24 h was between 4.9 and 5.2. Rola et al. [62] found *S. aureus* but no SEs in the small-scale production of raw milk cheese from cow’s milk in Poland. The cheese was heated to max. 32 °C, ripening time was two weeks. The *S. aureus* log(CFU) values in the formed cheese (80.8% positiv) were between <1 and 6.04, while in the ripened cheese (69.2% positiv) they were between <1 and 7.41.

Borelli et al. [63] examined Brazilian Canastra raw cow’s milk cheese at farmhouse level. The cheese was made according to artisanal procedures using natural starter cultures. Samples were taken on the fifth day of ripening (at least 60 days for commercialization), log(CFU) values ranged from <2 to 6.3. In another Brazilian study, Aguiar et al. [64] found log(CPS) values between 2.38 and 3.7 on the first day in traditional artisanal raw milk colonial cheese Diamante, they were free of SE. In Cuba, Martínez-Vasallo et al. [65] found between 4.0 and 6.9 log(CFU) in artisanal fresh cheese. The Canadien Food Inspection Agency [66] collected different types of raw milk cheese samples from national retail chains and local or regional grocery stores. The 1715 of 1723 domestic and imported cheeses samples were satisfactory (≤3 log(CFU)), 4 investigative (between 3 and ≤4 log(CFU), Portuguese St. Jorge cheese), and two unsatisfactory (>4 log(CFU), one Portuguese St. Jorge cheese and one domestic Cheddar cheese).

Zeinhom and Abed [67] examined Egyptian Kareish fresh soft homemade cheese produced from buffalo or cow’s raw milk. They found between <2 and 5.78 log(CFU) *S. aureus*. In Turkey, Kaan Tekinşen and Özdemir [68] studied the prevalence of *S. aureus* in unripened, semi hard, dry salted Van otlu cheese obtained from retail outlets in Van and Hakkari markets. The cheese is made from raw ewe’s, cow’s and/or goat’s milk without starter cultures, and with herbs, and is usually produced in small primitive dairies under poor hygienic conditions. All samples were *S. aureus* positiv and the values ranged from 2.48 to 7.15 log(CFU).

The data from this study can be best compared with the results from the Norwegian summer-farm facility. Process parameters and results look quite similar. The Polish data for the formed cheese were also comparable, as were the data of the Scottish cheeses, of both Brazilian cheeses Canastra and Diamante, and the Egyptian Kareish cheese, but detailed information is not available and comparison difficult. Cuban fresh cheese and Turkish Van otlu cheese appear to be more contaminated. The Canadian data cannot be compared as it is from the market and no details of the cheeses etc. were reported.

Differences in contamination levels were observed based on cheese type. In hard cheeses (HC), no *S. aureus* was detected, likely due to the high scalding temperatures used during production (52–53 °C). Soft and semi-hard cheeses (SHC) showed varied levels of *S. aureus*, with SE-positive samples being associated with low maximum applied temperatures (37–43.6 °C) and inadequate acidification. Fresh cheeses (FC) exhibited the highest prevalence of *S. aureus*, with 83.3% of samples testing positive. However, all *S. aureus* counts in fresh cheeses remained below the legal limits, and no SEs were detected.

Referring to *S. aureus*-negative samples, cheese type FC showed the smallest percentage (*n* = 1, 0.81% of all cheese samples or 16.7% of the six total FC samples), type SHC a medium percentage (*n* = 44, 35.77% of all cheese samples or 44.9% of the 98 total SHC samples), and type HC had the highest percentage (*n* = 19, 15.45% of all cheese samples or 100% of the 19 total HC samples).

In 32% of the samples, *S. aureus* was detectable, with log(CPS) values below 5. Cheese type FC showed the highest percentage (*n* = 5, 4.07% of all cheese samples or 83.3% of the six total FC samples), type SHC a medium percentage (*n* = 34, 27.64% of all cheese samples or 34.7% of the 98 total SHC samples), and type HC the smallest percentage (*n* = 0, 0% or 0% of the 19 total HC samples).

The remaining 16% of the samples showed log(CPS) values of more than 5, thus violating the legal requirements. There were only SHC samples (*n* = 20, 16.26% of all cheese samples or 20.4% of the 98 total SHC samples). In two cases SEs were detectable (1.6%) with corresponding log(CPS) values of 5.431 (sample no. 135, SHC) and >5.477 (sample no. 13, SHC).

Given these results, HC is confirmed to be safe regarding CPS values and the absence of SE. For FC, the picture was not good, as only one FC was *S. aureus* free and the remaining five had log(CPS) values between 2.21 and 4.79. There were no SE-positive samples. The most mixed picture was that of the largest group of cheese types, SHC. The *S. aureus* value ranged from undetectable (log(CPS) of <2) to a log(CPS) of >6, including the two SE-positive samples.

### 4.2. Factors Influencing the Growth of S. aureus and the Production of SEs

This study shows that several factors influence the growth of *S. aureus* and the production of SEs in cheese. One critical factor is the type of starter culture used. Bulk starter cultures promote immediate acidification, thereby reducing the risk of *S. aureus* growth. In contrast, direct starter cultures, if not preactivated, exhibit a lag phase, leading to higher pH levels and increased *S. aureus* proliferation.

Milk quality also plays a pivotal role. Contaminated milk, particularly that from cows with *S. aureus* mastitis (e.g., the GTB genotype), was identified as a critical risk factor. Goat milk posed a lower risk due to the absence of contagious GTB strains and the application of effective processing practices. Milk with somatic cell counts exceeding 150,000–200,000 cells/mL significantly increases the risk of contamination and should not be processed into raw milk cheese.

Temperature control was another key factor. Maximum applied temperatures below 48 °C were associated with higher *S. aureus* counts and increased SE production, whereas temperatures above 48 °C effectively reduced *S. aureus* growth.

When producing HC, producers applied a scalding temperature of 52 to 53 °C, which seems to be a very effective hurdle (see also Figure 3) and is in accordance with Bachmann and Spahr [69]. However, the influence of temperature on growth and SE production can vary depending on the strain and growth medium used [70]. While SE production can occur between 15 and 45 °C, the inactivation of *S. aureus* in foods is possible with specific heat treatment (e.g., in milk, complete inactivation occurred under conditions such as 57.2 °C/80 min, 60.0 °C/24 min, and 71.7 °C/0.14 min) [16]. Notably, the number of *S. aureus* cells is not always a reliable indicator for enterotoxins’ presence, because not all strains of *S. aureus* are enterotoxigenic or express enterotoxins under all conditions [6,28]. In cases in which *S. aureus* cells were destroyed through heat treatment, heat-resistant enterotoxins could remain biologically active and cause food poisoning. Notably, staphylococci’s heat resistance can increase as a_w_ decreases, but it begins to decline when a_w_ falls between 0.70 and 0.80 [71]. The producers of most other cheeses applied maximum temperatures of less than 48 °C, which support the growth of *S. aureus* (SE-positive sample no. 135: 37 °C; SE-positive sample no. 13: 43.6 °C). In these cases, other hurdles, such as pH and competing microbes, must be applied effectively. Starter cultures in fermented dairy products can inhibit the growth of *S. aureus* and the formation of SE, and their failure significantly increases the risk of contamination [72]. Bachmann and Spahr [69] mention the preservative effect of lactic acid bacteria, which can be partially attributed to the activation of the lactoperoxidase system and partially attributed to bacteriocins. Le Marc et al. [73] studied the kinetics of *S. aureus* as a function of the starter culture (lactic acid bacteria). *S. aureus* growth is generally inhibited when lactic acid bacteria reach a critical density. This is known as the Jameson effect [74]. Considering the two SE-positive cases in this study, in one case, the producer mentioned that the culture failed to acidify (sample no. 135, with a pH 6.52 after 2 h and “1 Bulk starter, Flora danica [mesophil]—culture can be 3–4 days old”; see also Appendix A). In the other case, the producer used an autochthonous culture (sample no. 13, with a pH 6.42 after 2 h, a pH before salting of 5.84, and an autochthonous starter; see also Appendix A). In both cases, we assume that *S. aureus* was able to grow because pH and competitive starter hurdles failed. Staphylococcal strains grow optimally at pH values between 6 and 7, though they can tolerate a range from 4 to 10. This range decreases when other growth parameters are non-optimal. Factors that influence *S. aureus* response to pH include initial count, nutrient availability, NaCl concentration, temperature, and atmosphere. Under anaerobic conditions, most strains fail to produce detectable SEs below pH 5.7 [16,55,75,76]. Therefore, in the case of the two SE-positive samples, neither the applied temperature nor the acidification was sufficiently effective to suppress the growth of *S. aureus*.

A significant difference in log(CPS) was detected between the bulk and direct starter culture types. The significant difference between these two starters is based on the higher log(CPS) values for direct types, including a trend toward a higher pH (see Section 3.4). Mullan [77] points out that the addition of traditional bulk starters results in a decrease of 0.1–0.2 pH units, which may have a significant effect, and that such cultures begin producing acid virtually immediately, whereas this decrease is missing with direct starters. Additionally, there may be a lag period before the culture commences growth and acid production.

Thus, the hurdles of pH and competing microbes strongly depend on the culture used. In addition to temperature, producers must strongly focus on the type of culture used and culture quality control as part of the daily routine. Producers of AOP-labelled cheeses must often use a given type of culture. In such cases, the use of a strongly acidifying culture that provides efficient competition is necessary.

The last hurdle considered in this study is ripening time, which may play a role in lowering *S. aureus* counts for long-ripened cheeses but will not influence SE formation, as this takes place in the first hours of cheese production [6,34]. The samples were taken before or directly after brining (see Section 2.1). Both SE-positive SHCs would have been ripened for four weeks, and therefore, the ripening time would have had only a limited effect on *S. aureus* counts. Bachmann and Spahr [69] mention that the survival of *S. aureus* depends on the type of starter culture used and the acidification rate. In their study, SHC (Tilsiter) showed a log(CPS) of approximately 4 after 30 days and approximately 2 after 60 days, and it was not detectable after 90 days. Pretto et al. [78] found log(CPS) values of approximately 3 after 30 days and approximately 2 after 60 days in Serrano SHC.

A second goal of the study was the identification of the key parameters responsible for enterotoxin production.

In addition to the technological hurdles considered, the quality of the raw milk is another key factor. *S. aureus* GTB is a highly contagious mastitis-causing pathogen in cows, whereas GTC and other genotypes cause sporadic, noncontagious mastitis [4]. In goat milk from northern Italian region of Lombardy, Romanò et al. [79] identified seven genotypic clusters of *S. aureus,* with CLR being the most common. Goat milk may therefore pose less of a risk than cow milk, as the former does not contain contagious GTB. Also, given the data on the goat milk cheeses in our study, we assume that goat milk cheeses pose less of a risk because of the absence of contagious GTB and effective hurdles being in place, as half of the producers applied maximum temperatures of >48 °C and used appropriate starter cultures, resulting in pH-before-salting values of <5.7. Even, so there was no significant difference in log(CPS) (see Section 3.3) between the cheeses made from cow and goat milk.

Two SHC producers used a mix of goat and cow milk. They applied maximum temperatures of 45 and 55 °C, used the bulk and bulk+direct starter types, and provided no pH-2-h values, but both measured a pH-before-salting of 5.3. The log(CPS) values were 4.230 and 5.079, respectively. There were no SEs in either case. Because there were no pH-2-h values, the reason for the high log(CPS) values remains open, as the acidification rate is decisive for the growth of *S. aureus*.

The key factors responsible for the enterotoxin production are therefore as follows: (I) *S.-aureus*-contaminated milk, especially mastitis-related milk because of a lack of *S. aureus* mastitis sanitation and regular monitoring via somatic cell count analysis (e.g., the California mastitis test (CMT); see also FACE [80], (II) maximum temperatures <48 °C, and (III) not using appropriate starter cultures (non-existent, non-functional, lacking quality control, or inappropriate for the type of cheese).

The third goal of the study was the identification of ways to improve food safety.

Producing SE-free raw milk cheese with negative or low *S. aureus* counts requires not only good manufacturing practice and applying effective hurdles but also a focus on milk quality [80]. Processing milk for human consumption obtained from animals showing clinical signs of udder disease is not permitted [52] and represents a risk given the aim of producing safe raw milk cheese. In the case of *S. aureus* GTB, it is highly contagiousness, and it can be assumed that by the end of the Alpine season, the entire herd is infected, and the milk is increasingly contaminated, especially given a sinking milk amount. Regular mastitis testing using screening tests (e.g., the California mastitis test) or somatic cell count analysis, even if there are no visible symptoms, is therefore essential to guarantee good milk quality. However, the somatic cell count requirement of ≤400,000/mL for raw cow milk, in accordance with Regulation (EC) No. 853/2004, is not sufficient for processing raw milk products. It is recommended that milk with individual cell counts from 150,000 to 200,000 per ml not be processed into raw milk cheese [47,81,82].

In this study, we did not ask producers about their experiences in cheese making, specific hygiene knowledge, hygiene management on the Alps and working experiences under Alpine conditions. Therefore, these factors could not be evaluated. Consequently, it is necessary to use robust recipes and production environments to avoid any negative influence on safety and quality through small changes in production management or climate.

Our results show that with functioning hurdles, the safe production of various types of artisanal raw milk cheese is possible under elementary conditions. Critical factors are a sufficient high temperature step and a good quality-controlled working starter culture.

### 4.3. Hurdle Index as a Tool for Risk Assessment

The HI was developed as a simple tool via which to assess the effectiveness of hurdle technology in the production of artisanal raw milk cheese regarding *S. aureus* and SE formation. This index incorporates key parameters, including maximum temperature; pH; FV, representing the activity of competing microorganisms; and ripening time. By integrating these factors, the HI enables the identification of high-risk production environments by evaluating whether the applied hurdles are sufficient to inhibit *S. aureus* growth and prevent the production of SEs.

The graphically chosen HI of 35 should reliably exclude the SE positive samples and as many samples as possible with log(CPS) values ≥5, without excluding the producers who theoretically and effectively produce uncontaminated cheese under the conditions of an HI of 35. An HI of 35 (see also Table 1) corresponds technologically to an FV of 1, a maximum temperature of 49 °C (>48 °C, see Figure 3) and a ripening time of 4 weeks (log(CPS) <5 [69]), and a pH 5.6 (<5.7 [16,55,75,76]). Under these technological conditions, considerable *S. aureus* growth and SE formation should not occur. Beside excluding all SE positive samples, a closer look at the distribution of the log(CPS) data reveals a good differentiation of the log(CPS) values ≥5 between samples with HI ≥35 and samples with HI <35 (HI ≥35 excludes 80% of samples with log(CPS ≥5)).

Key observations obtained from the study highlight the relationship between production parameters and HI scores. SE-positive samples, such as samples 13 and 135, were associated with low maximum temperatures (37–43.6 °C) and critical pH levels above 5.7, resulting in low HI scores of 22.70 and 29.86, respectively. Conversely, safer production environments were characterized by higher maximum temperatures (>48 °C) and lower pH values (<5.7), which corresponded to higher HI scores, indicating the reduced risk of *S. aureus* growth and SE formation.

As the samples in this study were taken at the time of the highest CPS counts, the HI can only be verified to a limited extent. While ripening time was initially considered a significant factor, its role was found to be overestimated in some cases. Although ripening time is important in reducing *S. aureus* contamination over time, and has therefore a major influence on the assessment of cheese at the time of market maturity, it has limited influence on SE production, which primarily occurs during the early stages of cheese production. This finding underscores the importance of focusing on immediate production parameters, such as temperature and pH, to enhance the safety of artisanal raw milk cheese.

The HI must be tested and validated in other studies with larger data sets, including whether it is suitable in this form for the assessment of other microbiological risks (in some cases with completely different contamination pathways). Various approaches are conceivable for this, with production conditions that are as balanced or uniform as possible (cheese type, maturing time, milk type, etc.), which should, if possible, also take into account the raw milk quality with regard to mastitis (CMT or better still CPS counts). Furthermore, the quality of the culture, the pH value and the *S. aureus* counts during ripening up to sale should also be documented.

Finally, based on this study, safe artisanal raw milk products can be produced if the raw milk quality is high (no or few *S. aureus*), the maximum temperature applied is ≥48 °C, and the starter culture is suitable and able to rapidly reduce the pH. For hard cheese this means, it is essential to maintain a scalding temperature of ≥48 °C in combination with a sufficient acidification rate, whereas for fresh, soft, and semi-hard cheese beside the sufficient acidification rate a high raw milk quality is a fundamental requirement.

This finding is also described in specific guidelines regarding other pathogens (Table 2).

## 5. Conclusions

In this study, we investigated *S. aureus* contamination and enterotoxin production in raw milk cheeses from the Alpine regions of Austria, Italy, and Switzerland. We focused on identifying the key factors influencing contamination, toxin production, and way to improve food safety in artisanal raw milk cheese production. Additionally, this study introduced the HI to assess the effectiveness of technological and environmental hurdles in ensuring safe production. The HI value of 35 identified by our dataset to discriminate safe cheeses, although it needs to be validated on a larger number of data, provides a simple and effective method for assessing the potential risk of the presence of SE. This study demonstrates that safe artisanal raw milk cheese production is achievable under elementary conditions by applying effective hurdles, including high scalding temperatures or thermization, quality starter cultures, the measurement of their activity via the pH 2 h after moulding, and robust milk quality management. The introduction of the HI provides a promising tool for assessing and improving safety in raw milk cheese production. While ripening time plays a limited role in SE prevention, it remains important in reducing *S. aureus* contamination over time. By addressing these factors, producers can ensure the safety, in terms of meeting legal requirements, and quality of raw milk cheeses while preserving their traditional characteristics.

## Figures and Tables

**Figure 1 foods-14-02176-f001:**
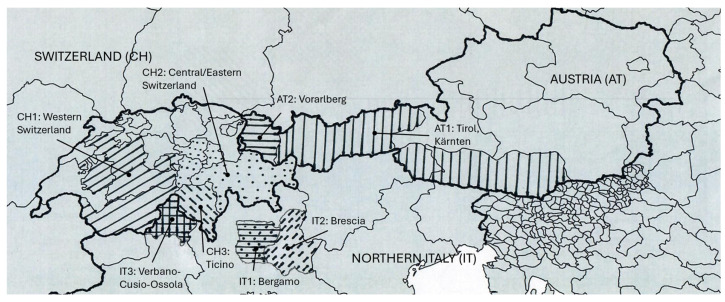
Sampling zones in the Alpine regions of Austria (AT1: Tirol-Kärnten, AT2: Vorarlberg), Italy (IT1: Bergamo, IT2: Brescia, IT3: Verbano-Cusio-Ossola), and Switzerland (CH1: Western Switzerland, CH2: Central/Eastern Switzerland, CH3: Ticino). See also the region groups in Appendix A.

**Figure 2 foods-14-02176-f002:**
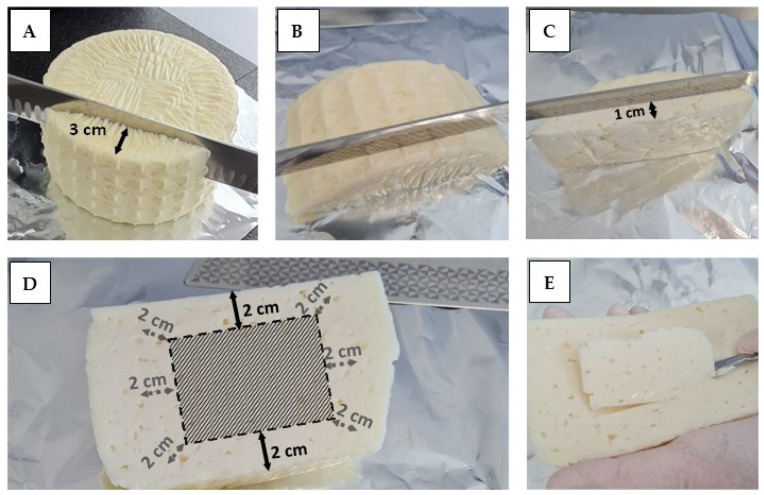
Sampling instructions for manufacturers and taking the test quantity for the enumeration of coagulase-positive staphylococci (CPS) in cheese before ripening at the lab site. (**A**): cutting off a sample approximately 3 cm from the edge and cooled transport to the lab site within 24 h, (**B**): aseptic cutting off the upper surface, (**C**): aseptic cutting of a 1-cm slice (distance from the cheese inside), (**D**): aseptically taking the test quantity for microbiological analysis at a distance of approximately 2 cm to all edges, (**E**): test quantity.

**Figure 3 foods-14-02176-f003:**
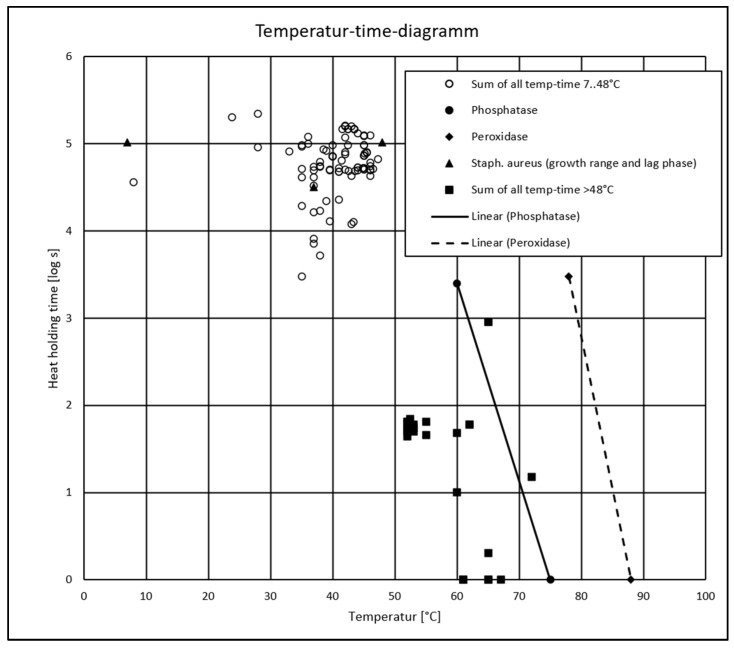
Heat load (heat treatment of the milk as well as holding/scalding temperatures) of the 126 samples, shown as maximum applied temperature (°C) versus holding time (log s); see Appendix A). Phosphatase and peroxidase inactivation times are shown as lines, and *S. aureus* growth range and lag phase are shown as triangles (To estimate the critical range of S. aureus growth (temperature-time), the data obtained from Taitini [55] (*S. aureus* grows between 7 and 48 °C, with temperature being optimal at around 37 °C) and Lindquist [56] (the mean lag times ranged from 8. 8 to 19.5 h for strain S30 and from 12.2 to 28.7 h for strain S119) are used. There was minimum growth from 7 °C and maximal growth up to 48 °C, with the largest lag time of 28.7 h, and optimal growth at 37 °C, with the smallest lag time of 8.8 h. The growth of S. aureus can be expected from these timepoints on at the latest if no other limiting factors are present). Three samples (no. 31, 90, and 92, partly overlapping) were subjected to time-temperature intervals corresponding to pasteurization (left of the phosphatase inactivation line). These samples were excluded from the statistical evaluation.

**Figure 4 foods-14-02176-f004:**
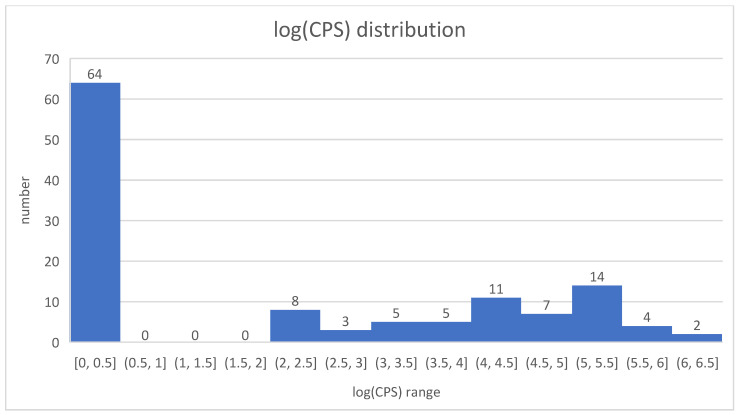
Distribution of the log(CPS) values (coagulase-positive staphylococci (CPS)), *n* = 123, 64 (52%) <0.5, 11 (8.9%) between 2 and 3, 10 (8.1%) between 3 and 4, 18 (14.6%) between 4 and 5, 18 (14.6%) between 5 and 6, and two (1.6%) between 6 and 6.5.

**Figure 5 foods-14-02176-f005:**
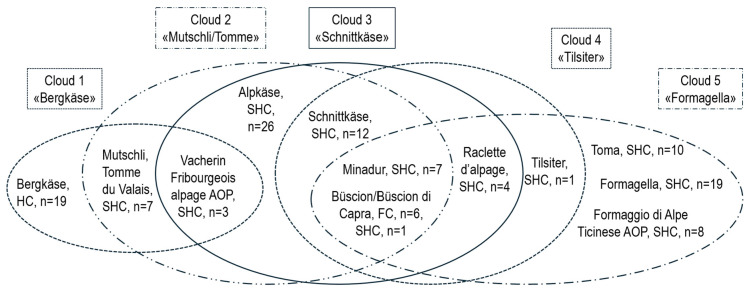
Cheese variety Clouds 1–5. Each of the partially overlapping clouds contains cheese varieties that did not show significant differences in log(CPS), coagulase-positive staphylococci (CPS), *n* = 123. Fresh cheese (FC), hard cheese (HC), semi-hard cheese (SHC), AOP/POD (Product of Designated Origin).

**Figure 6 foods-14-02176-f006:**
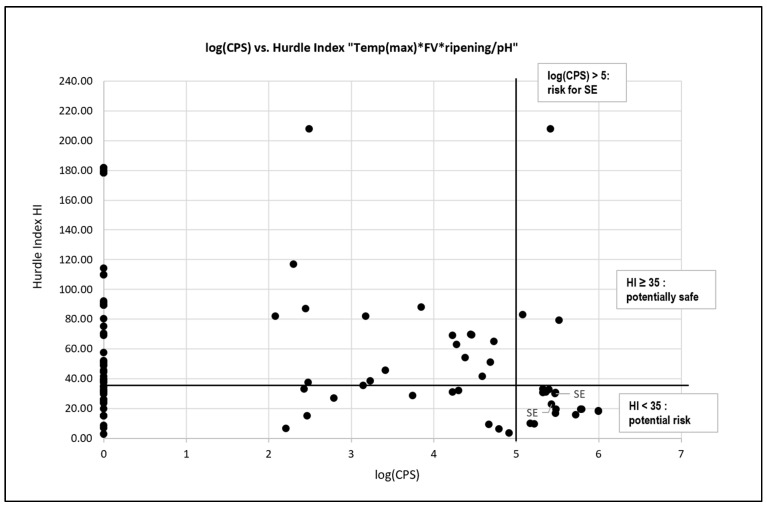
log(CPS) values, coagulase-positive staphylococci (CPS), versus Hurdle Index (HI). A HI of 35 appears to be a reasonable basis for evaluating raw milk cheeses with respect to the CPS or Staphylococcal Enterotoxin (SE) risk. Flora Value (FV).

**Figure 7 foods-14-02176-f007:**
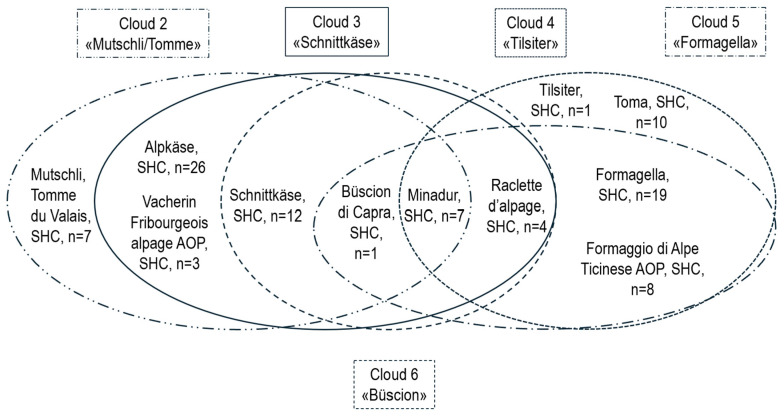
Cheese variety Clouds 2–6. Each of the partially overlapping clouds contains cheese varieties that did not show significant differences in log(CPS) (coagulase-positive staphylococci (CPS), semi-hard cheese (SHC) only, *n* = 98). AOP/POD (Product of Designated Origin).

**Table 1 foods-14-02176-t001:** List of cases from this study, their technological parameters, Flora Values (FV), the calculated Hurdle Indices (HI) and measured log(CPS) values, Coagulase-positive staphylococci (CPS), for further evaluation of the HI. The two Staphylococcal Enterotoxin (SE) positive samples 13 and 135 are indicated in bold.

HI	log(CPS) [-]	FV [-]	Max. Temp. [°C]	Ripening [w]	pH [-]	Sample no.	Comment
6.06	4.972	1	28.00	1	4.62	79	(a)
6.56	2.206	1	28.00	1	4.27	78	(a)
9.59	5.223	0.1	37.00	14	5.40	4	(b)
9.77	5.170	0.1	37.00	14	5.30	1	(b)
15.76	5.724	0.5	39.00	4	4.95	15	(b)
**22.70**	**5.431**	**1**	**37.00**	**4**	**6.52**	**135**	(c)
29.52	0	1	62.00	2	4.20	84	(d)
**29.86**	**>5.477**	**1**	**43.60**	**4**	**5.84**	**13**	(c)
30.71	>5.477	1	41.00	4	5.34	11	(e)
34.44	0	1	62.00	3	5.40	83	(d)
34.72	0	1	38.00	5.5	6.02	129	(f)
35	0	1	49	4	5.6		
57.27	0	1	45.10	8	6.3	43	(g)
79.00	5.516	1	39.50	12	6	103	(h)
83.02	5.079	1	55.00	8	5.30	80	(i)
114.17	0	1	42.00	14	5.15	34	(j)
207.69	2.493	1	45.00	24	5.20	86	(k)
207.69	5.415	1	45.00	24	5.20	87	(k)

**Table 2 foods-14-02176-t002:** Comparison of recommendations in various guidelines for artisanal raw milk cheeses from Alps. California mastitis test (CMT), Somatic Cell Count (SCC), Fresh cheese (FC), hard cheese (HC), semi-hard cheese (SHC).

Parameter/Guideline		[48]	[82]	[47]	[81]
milk quality of individual dairy animal (either CMT or SCC)	CMT	-	inconspicuous	inconspicuous	inconspicuous
	SCC	-	<150,000/mL	<150,000/mL	<200,000/mL
	frequency	-	regular testing, e.g., every month	regular testing, e.g., 1–2x/month	regular testing, e.g., every 14 days
Scalding temperature [°C]	HC	≥48	≥52	50–57	50-53
	SHC	-	typically <46, eventually thermization	40–48	-
	FC	-	only with thermized milk	only with pasteurized milk	-
Acidification rate pH [-]	HC	<6.2 (after 2 h)	-	<6.2 (after 2 h)	<6.2 (after 2 h)
	SHC	<6.0 (after 2 h)	-	≤5.4 (before salting)	<6.0 (after 2 h)
	FC	<5.0	<4.5 (after 2 h)	<5.0 (after cutting)	-
Ripening time [d]	HC	-	120	>120	-
	SHC	>60 d	>60 d	-	-
	FC	-	-	-	-

## Data Availability

The original contributions presented in the study are included in the article/Appendix A, further inquiries can be directed to the corresponding author.

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
