# Peer review of "Critical Factors Affecting the Prevalence of Staphylococcus aureus and Staphylococcal Enterotoxins in Raw Milk Cheese in the Alpine Region of Austria, Italy, and Switzerland"

_foods, 2025, doi:10.3390/foods14132176_

Round 1
Reviewer 1 Report
Comments and Suggestions for Authors
The article is very interesting, because its goal is to provide not only the overview of S. aureus and its enterotoxins in raw milk cheeses in the Alpine regions in Europe, namely Austria, Italy, and Switzerland), but also identify the key parameters which are responsible for enterotoxin production. The ways to improve food safety in this type of cheeses was also included to the experiment. It is generally known, that Alpine regions (Figure 1) in all evaluated countries is rich for the mountain pastures and excellent conditions for manufacturing dairy products, mostly different type of cheeses made from raw milk Cow, goat). That is why the identification of potential biological hazard is very important. Authors of the article used in their study exact methods for isolation and identification of S. aureus from cheese samples (126 cheese samples) taken from different parts of manufacturing process. Sampling of cheeses id described very detailed and the Figure 2 very precisely described the instructions for cheese manufacturers. It is very important. Also used questionnaire survey on manufacturing conditions helps to authors in the identification of stated key parameters.
I agree with author’s statement that the hurdle index introduced in this paper is a promising tool for assessing and improving safety in the production of cheeses made from raw milk and can be another part of improving safety in raw milk cheese production.
I have following comments to the authors in the part of “References”:
- Citations No. 3, 5, 7, 8, 25, 64, 69 have a letter with a dot at the end, the other citations lack a dot at the end.
- Citation No. 9 - There is a period after the year 2009, and a comma elsewhere. Which is correct?
- Citations No. 10, 11 – Staphylococcus aureus is not written in italic.
- Citation No. 38. UNESCO – at the end of the citations in brackets is written: “assessed 31. August 203). Is that spelled correctly?
- Citation No. 41. The name of the journal “Revue de Géographie alpine” .Is that spelled correctly?
Reviewer 2 Report
Comments and Suggestions for Authors
The study provides a valuable overview of Staphylococcus aureus and enterotoxin contamination in Alpine raw milk cheeses, integrating microbiological data with technological parameters. The introduction of the Hurdle Index (HI) as a novel tool for risk assessment is innovative, and the focus on traditional cheese-making practices in transhumant systems adds contextual relevance to food safety challenges in marginalized agricultural regions. The multi-country sampling (Austria, Italy, Switzerland) and characterization of genotypes contribute to understanding regional contamination patterns. Some issues still need to be clarified.
- Clarity in Sampling Strategy and Regional Representation.
The study combines cheese types and regions but does not explicitly justify the grouping or address potential biases in sample distribution. How were participating producers recruited, and what measures were taken to ensure representativeness of artisanal practices?
- Genotyping and Virulence Gene Analysis
While the study mentions GTB genotypes are linked to contagious mastitis, it lacks detailed data on the prevalence of specific enterotoxin genes (sea, seb, etc.) across genotypes. For example, do GTB strains in Swiss cheeses (CH3) consistently carry classical SE genes, and how does this correlate with SE detection in SHC samples? The use of ribosomal spacer PCR for genotyping is noted, but newer methods (e.g., MLST, whole-genome sequencing) could provide deeper insights into strain diversity and virulence potential. Were these technologies considered, and what are the limitations of the current approach?
- Validation and Limitations of the Hurdle Index (HI)
The HI formula combines temperature, ripening time, pH, and starter quality, but ripening time’s relevance to SE production (which occurs early) is questionable. Should the HI be split into two models: one for SE risk (excluding ripening) and one for general S. aureus contamination?
The threshold HI = 35 is derived from a small dataset (123 samples). How would the authors validate this in larger cohorts, especially for cheeses with non-standard parameters (e.g., goat milk, short-ripening FC)?
- Starter Culture Characterization and Acidification Kinetics
Bulk starters are shown to acidify faster than direct starters, but the study does not quantify lactic acid production rates or bacterial counts over time. For instance, in SE-positive sample 135 (bulk starter, pH 6.52), why did acidification fail despite using a commercial culture? Was the starter’s viability or age (e.g., "3–4 days old") a factor?
"Natural" starters are categorized as having a Flora Value (FV) of 0.5, but their microbial composition (e.g., autochthonous LAB strains) is not characterized. How does this affect their competitive efficacy against S. aureus?
- Milk Quality Metrics and Mastitis Management
The study emphasizes mastitis as a risk factor but does not report somatic cell count (SCC) data for individual farms. Were SCC thresholds (e.g., >150,000 cells/mL) used to exclude high-risk milk, and how did SCC correlate with GTB prevalence in raw milk?
In Table S2a, some milk storage temperatures exceed EC regulations (e.g., 18°C for 12 hours). How did such practices influence S. aureus growth during storage, and were they documented as risk factors in the statistical analysis?
- Thermization vs. Raw Milk: Efficacy and Safety
Thermized milk (phosphatase-positive) is included, but the study does not compare its CPS counts to raw milk. For example, in CH3 (Ticino), was thermization consistently applied at ≥48°C, and did it reduce S. aureus counts more effectively than in non-thermized samples? Pasteurized samples were excluded, but what criteria defined "thermized" vs. "raw" milk in the study, and how was heat treatment intensity quantified (e.g., time-temperature profiles)?
- Goat Milk Safety: Genomic and Ecological Insights
Goat milk cheeses lacked GTB and classical SEs, but the study suggests genetic differences in S. aureus strains. Were goat milk isolates genotyped (e.g., CLR clusters), and are there ecological factors (e.g., pasture microbiome, host immunity) that reduce contagious mastitis in goats?
Better to improve the quality of figures and tables.
Reviewer 3 Report
Comments and Suggestions for Authors
some minor corrections as follows:
Line 44: Please correct "thee" to "the".
Lines 68–69: I suggest adding Commission Regulation (EC) No. 2073/2005 to the list of references.
Line 108: The abbreviation SEH is not explained with its full term.
Lines 816, 823, 828: Please italicize Staphylococcus aureus.
Line 894: Please correct "Kummel" to "Kümmel".
Line 928: Reference number 35 is missing from the main text.
Please remove the underlining from references 21–24 and 43.
Please delete the extra spaces between references in the reference list.
Some reference numbers are displayed in blue, while others are black. Please standardize the color formatting.
Each figure and table should be self-explanatory and not rely on the main text for clarification. Please, write all abbreviations present in them using full terms.
Reviewer 4 Report
Comments and Suggestions for Authors
This is a comprehensive study of artisanal raw milk cheeses in the Alpes. The hurdle index is a useful tool to assess the risk of S. aureus contamination and possible SE production. However, who would adopt or enforce adoption of such an index so that producers follow procedures to reduce the risk of SE in their cheeses. Although the study did not look at other potential contaminants like Salmonella or measure Enterobacteriaceae, there could some comment on increased risk from these in the Discussion or Conclusion. other comments below.
Not all readers will know about AOP Appellation d’Origine Protégée (Protected Designation of Origin).
Define what you mean by Clouds.
You measure the HI but what is the actual risk to the consumers? Have there been aureus SE outbreaks from any of these cheeses or recalls? What about from other pathogens like Salmonella or Listeria?
What is the origin of S. aureus regardless of strain? Apart from cow (or other animal) milk what about human S. aureus sources; also, starter contamination? I note late summer the risk of mastitic milk increases - discuss a bit more and whether any prevention/controls might help.
Under 3.5 there is a table without number or title. Correct.
The conclusion in this sentence has been said at least 3 times in the manuscript. “Although ripening time is important in reducing S. aureus contamination over time, it has limited influence on SE production”
What is the shelf-life of HC, SHC and FC bearing in mind shoppers often store longer than recommended. Does that increase the risk in any of these cheeses, or has growth of S. aureus stopped by then (what about risk of other pathogens like Salmonella or Listeria?)
Check spelling of meso-/themopilic culture in Table S2b. Why is morning vs. morning/evening important for adding starter?
Why S2a, S2b, S2c instead of S2, S3, S4?
Reviewer 5 Report
Comments and Suggestions for Authors
This manuscript provides a comprehensive investigation into the prevalence of Staphylococcus aureus and its enterotoxins in raw milk cheeses produced in the Alpine regions of Austria, Italy, and Switzerland. It addresses a significant food safety concern in artisanal cheese production, particularly in the context of traditional transhumance practices. The introduction of the Hurdle Index (HI) as a novel tool for assessing the effectiveness of hurdle technologies is a notable contribution. The study is well-designed, with robust sampling and analytical methods, and the results are clearly presented. However, there are areas where clarity, methodological detail, and discussion could be improved to enhance the manuscript’s impact. The manuscript is suitable after addressing the following major and minor revisions.
The threshold of HI = 35 is proposed as a safety benchmark, but its derivation is unclear. The manuscript states it “appears to be a reasonable basis” (line 341), but this is not sufficiently rigorous. Describe how this threshold was determined (e.g., statistical analysis, receiver operating characteristic curves) and discuss its limitations.
The discussion acknowledges that ripening time may have limited influence on SE production (lines 411–413), yet it is included in the HI formula. Consider revising the HI formula for SE risk assessment to exclude ripening time or justify its inclusion more clearly.
The manuscript does not discuss potential sampling biases. For example, were manufacturers randomly selected, or was participation voluntary? Voluntary participation could bias the sample toward producers with better practices. Clarify the sampling strategy and discuss its implications.
The questionnaire on manufacturing conditions (lines 213–219) is critical for interpreting results, but its design (e.g., question types, response rates) is not described. Provide details on the questionnaire’s structure, validation, and how incomplete responses were handled.
The discussion could be strengthened by comparing the findings to other studies on S. aureus in raw milk cheeses outside the Alpine region. For example, how do the prevalence rates and risk factors compare to those in other European countries or North America?
The chi-squared test is mentioned (line 260) but not clearly linked to specific results. Specify where it was used and report the results.
Line 44: “thee culling” should be “the culling”.
Line 237: “1 L of α-cyano-4-hydroxycinnamic acid” is incorrect; it should be “1 μL” (microliters). Correct this typo.
In Table 1, “SSC” is used instead of “SCC”
Line 43: “unsuitability of the milk and dairy products” is vague. Rephrase for clarity, e.g., “reduced suitability of milk for dairy product manufacturing.”
Line 141: “allowed individuals to survive as mountains farmers” should be “mountain farmers”
“pH of 6.42” should be “pH 6.42” for consistency with scientific notation. Follow the instruction for all the pH written in the manuscript.
Line 292: “Chapter 2.1” is referenced, but the manuscript does not use chapter headings. Replace with “Section 2.1”
Round 2
Reviewer 2 Report
Comments and Suggestions for Authors
Can be accepted.
Reviewer 5 Report
Comments and Suggestions for Authors
NA